# Initial Outcomes of Physician-Modified Inner-Branched Endovascular Repair for Distal Aortic Arch Aneurysm

**DOI:** 10.3390/jcm14010039

**Published:** 2024-12-25

**Authors:** Shingo Tsushima, Tsuyoshi Shibata, Yutaka Iba, Tomohiro Nakajima, Junji Nakazawa, Shuhei Miura, Ayaka Arihara, Kei Mukawa, Yu Iwashiro, Nobuyoshi Kawaharada

**Affiliations:** Department of Cardiovascular Surgery, Sapporo Medical University, Sapporo 060-8556, Japan; s.tsushima@sapmed.ac.jp (S.T.); iba-yu@sapmed.ac.jp (Y.I.); t.nakajima@sapmed.ac.jp (T.N.); bankara@sapmed.ac.jp (J.N.); smiura@sapmed.ac.jp (S.M.); a.arihara@sapmed.ac.jp (A.A.); k.mukawa@sapmed.ac.jp (K.M.); rfsmu0726@gmail.com (Y.I.); nobuyosh@sapmed.ac.jp (N.K.)

**Keywords:** physician-modified inner-branched endovascular repair, distal aortic arch aneurysm, initial outcomes

## Abstract

**Background/Objectives:** Few clinical studies have reported on physician-modified inner-branched endovascular repair (PMiBEVAR) for aortic arch aneurysm. Herein, we evaluate the outcomes of proximal landing zone 2 PMiBEVAR. **Methods:** This retrospective study analyzed data from six patients who underwent zone 2 PMiBEVAR for aortic arch aneurysms at a single center between October 2021 and June 2024. The outcomes were in-hospital mortality and postoperative complications. **Results:** The median follow-up period was 12.5 (7.3–25) months. Males constituted four out of six cases and females constituted two out of six cases. The patients had a median age of 78.5 (76.5–79.0) years, and the aneurysm diameter was 56 (50–61) mm. Technical success was achieved in 5/6 (83.3%) cases. The median modification and operative times were 56 (45–60) min and 92 (79–308), respectively. Postoperatively, delayed type Ia endoleak and vascular access-site pseudoaneurysm occurred in one patient each. However, no patients experienced other complications. The median hospital stay was 10 (7–41) days, and no deaths or reinterventions occurred after 30 days. During the post-discharge follow-up, the aneurysm diameter remained “unchanged” in four patients, including the one with delayed type Ia endoleak, while two patients experienced “shrinkage”. **Conclusions:** PMiBEVAR for distal aortic arch aneurysm might be effective in improving postoperative outcomes.

## 1. Introduction

Although open repair is the “gold standard” for treating aortic arch diseases, thoracic endovascular aortic repair (TEVAR) has been shown to be a safe and effective alternative for patients with aortic arch aneurysms who have a high surgical risk. The endovascular and hybrid repair of the aortic arch has seen a rise in novel technologies and the literature over the last decade [1]. The procedure, which involves proximal landing zone 2, requires options such as debranching, fenestration, or the chimney technique to ensure the perfusion of the left subclavian artery (LSCA). However, debranching requires an extra-anatomical bypass [2], and aneurysms involving supra-aortic branches are not optimal candidates for the fenestration of the endoprosthesis [3]. Moreover, the chimney technique is associated with a high risk of gutter endoleaks and stent-graft migration [4]. Although frequently requiring debranching to maintain the patency of LSCA, endovascular arch repair is an attractive alternative procedure for high-surgical-risk patients. The fenestrated endograft repair of aortic arch disease has a high technical success rate (95%); however, a high number of secondary interventions were needed [5]. On the other hand, zone 2 TEVAR using the chimney technique has a high risk of type Ia endoleaks [4].

Recently, the use of physician-modified inner-branched endovascular repair (PMiBEVAR) for aortic arch aneurysm has been increasingly reported. For example, Zhang et al. [6] described a case of successful total endovascular repair of an aortic arch aneurysm using a physician-modified endograft with triple inner branches. Although acceptable initial outcomes after PMiBEVAR for thoracoabdominal aortic aneurysm have been reported [7], clinical studies regarding PMiBEVAR for aortic arch aneurysms are currently limited. Based on this background, this study aimed to evaluate the initial outcomes of six cases in which PMiBEVAR was performed at proximal landing zone 2. We hypothesized that PMiBEVAR was a minimally invasive and acceptable procedure when compared to previous techniques.

## 2. Materials and Methods

At our institution, we perform PMiBEVAR, using an inner branch and bridging stent-graft to ensure perfusion of the LSCA. This retrospective study analyzed data from six patients who underwent zone 2 PMiBEVAR for distal aortic arch aneurysms at a single academic medical center between October 2021 and June 2024.

The surgical criteria in the center adhered to global standards [8], with interventions recommended for aneurysms meeting size-specific criteria, those presenting symptoms including rupture, or those exhibiting rapid size expansion. Ultimately, decision-making is based on the JCS/JSCVS/JATS/JSVS 2020 Guideline [9], combined with a detailed assessment of the aortic anatomy and a thorough evaluation of the patient’s comorbidities. Each center established a multidisciplinary team to evaluate patients’ overall health conditions for elective surgery, categorizing patient comorbidities.

Patient eligibility for PMiBEVAR followed specific criteria: (1) an American Society of Anesthesiologists (ASA) score [10] of ≥3 points or inapplicability of conventional open repair due to previous aortic open surgery or severe comorbidities, and (2) a proximal landing zone (the distance between the proximal end and the ostium of the LSCA) of at least 15 mm for distal aortic arch aneurysm. Patients underwent baseline clinical examinations, laboratory tests, and imaging (computed tomography; CT), with subsequent follow-ups at 6–8 weeks, 6 months, and 12 months postoperatively, followed by annual assessments.

Modification of the stent-grafts was performed by two cardiovascular surgeons. The thoracic stent-graft was partially unloaded using a delivery system and the proximal barb was removed. Based on measurements from preoperative CT Angiography (CTA), electrocautery was used to create a fenestration in a predetermined location on the LSCA to match the curve of the greater curvature of the aortic arch. The inner branch of the LSCA was prepared using a Viabahn^®^ stent-graft (WL Gore & Associates, Flagstaff, AZ, USA) cut to 10–15 mm and securely affixed to the fenestration using running 5-0 Ethibond^®^ or Prolene^®^ sutures (Ethicon, Inc., Bridgewater, NJ, USA), using an Azur^®^ coil (Terumo Medical, Tokyo, Japan) or ONE Snare^®^ (Merit Medical, Tokyo, Japan) as a marker. To enable the approximation of the direction of physiologic blood flow, the inner branch was not fixed to the main device. Subsequently, the modified stent-graft was manually resheathed while tightening the graft with vascular tape. The physician-modified stent-graft was then advanced into the aortic arch via the femoral artery. After unsheathing to open the fenestration, the bridging stent-graft was inserted into the inner branch of the LSCA from the left upper limb, while securing the main graft to avoid rotation during delivery.

The primary endpoints of this study were technical success, in-hospital mortality, and postoperative complications. Imitating other PMiBEVAR studies [7], technical success was defined on an intention-to-treat basis as successful endovascular access and deployment of all devices, successful catheterization and stenting of the LSCA, and no target vessel occlusion on control angiogram. Postoperative complications were defined using a composite endpoint following previous reports [2,4], including all-cause mortality, endoleak, acute renal failure (>50% decrease in estimated glomerular filtration rate), spinal cord ischemia (non-ambulatory, Spinal Cord Injury scale grade 3a–3c) [11], and stroke. We routinely performed postoperative CTA to check whether any endoleaks occurred within 1 week. After discharge, we continued follow-up using plain CT at 6–8 weeks, 6 months, and 12 months postoperatively and annually thereafter. Decisions regarding reintervention were made based on the following factors: aneurysm growth of >5 mm when compared with the preoperative measurement, the presence or absence of type I or type III endoleaks, graft migration, graft occlusion, rupture, and infection following the guideline [9]. If endovascular reintervention was not suitable due to anatomical problems or reintervention failure, conversion to open surgery was performed. Aortic remodeling was classified as follows: shrinkage (a diameter reduction of ≥5 mm), no change (a diameter reduction or increase of <5 mm), and enlargement (a diameter increase of ≥5 mm).

### 2.1. Statistical Analysis

All statistical analyses were performed using JMP Pro version 17 (SAS Institute Inc., Cary, NC, USA). Continuous variables are expressed as mean ± standard deviation when normally distributed and median (interquartile range [25–75%]) when non-normally distributed.

### 2.2. Ethical Approval

This study was conducted in accordance with the principles of the Declaration of Helsinki and approved by the Institutional Review Board of our university (no. 362-25). The requirement for informed consent was waived as no identifiable patient information was used.

## 3. Results

This study included six consecutive patients who were treated with PMiBEVAR at a single academic medical center between October 2021 and June 2024. The study flow chart is shown in Figure 1. The preoperative characteristics of the patients are listed in Table 1. The median age was 78.5 (76.5–79.0) years, and four out of six (66.7%) patients were male. The median aneurysm size was 56 (50–61) mm, and all patients had hypertension. Regarding the histological components of the aorta, the proportions of true aneurysms, pseudoaneurysms, and dissecting aneurysm were 83.3%, 0%, and 16.7%, respectively. Referring to the ASA class, surgical risks tended to be higher. Additionally, half of the cases had previously undergone aortic surgery.

Regarding the procedural details (Table 2), the median modification and operative times were 56 (45–60) min and 92 (79–308) min, respectively. The Zenith Alpha™ thoracic stent-graft (Cook Medical, Bloomington, IN, USA) was used in most patients, whereas the Relay^®^Plus thoracic stent-graft (Terumo Aortic, Sunrise, FL, USA) was used in only one patient. In all patients, the inner branch of the LSCA was prepared using a Viabahn^®^ stent-graft (WL Gore & Associates, Flagstaff, AZ, USA), which was securely affixed to the fenestration (Figure 2A). The bridging stent-grafts used for the LSCA were as follows: LifeStream™ (Becton, Dickinson and Bard Company, Tempe, AZ, USA), Viabahn^®^ VBX (WL Gore & Associates, Flagstaff, AZ, USA), and Viabahn^®^ (WL Gore & Associates). In all cases, endovascular treatment was achieved through a percutaneous approach to the access vessel, and the bridging stent-graft was inserted into the inner branch of the LSCA (Figure 2B). Technical success was achieved in five out of six (83.3%) cases. Open abdominal surgery was required in one patient due to access vessel bleeding. Figure 3 shows the CT images obtained pre- and postoperatively. Device selections for each case are shown in Table 3. Table 4 shows the postoperative outcomes of the patients. In total, one patient developed a delayed type Ia endoleak and another experienced a vascular access-site complication. However, no patients experienced stroke, spinal cord ischemia, or any other complications. The median length of hospital stay was 10 (7–41) days, with no deaths occurring during hospitalization. Table 5 shows the post-discharge follow-up outcomes. The median follow-up period was 12.5 (7.3–25) months. Notably, the aneurysm diameter remained “unchanged” in four patients, including the patient with a delayed type Ia endoleak. The two other patients experienced aneurysm diameter shrinkage. Fortunately, no cases of aneurysm diameter enlargement or endoleak were observed. Of the six patients, one died due to respiratory failure. Table 6 shows the data for each case. The cases were listed in order of the dates on which PMiBEVAR was performed. Only the first case resulted in a technical failure. In the cases after the second, the operative time and the postoperative length of stay were shorter.

## 4. Discussion

Our study findings show the feasibility of PMiBEVAR for distal aortic arch aneurysm. TEVAR at proximal landing zone 2 requires LSCA revascularization. In this study, we considered PMiBEVAR to be a minimally invasive and acceptable procedure when compared to previous techniques such as debranching, fenestration, or the chimney technique, as all cases were performed using percutaneous approach, with a shorter operative time and no in-hospital deaths. In a previous study on the treatment of aortic arch aneurysm, no significant difference in 3-year mortality was found between the open aortic repair and hybrid TEVAR with debranching techniques [12]. Conversely, Squiers et al. [2] compared outcomes between a surgical debranching group and branched endograft group in zone 2 TEVAR and found that the procedural time was significantly longer for the surgical debranching group. Moreover, Tsilimparis et al. [13] performed a retrospective review of fenestrated endoprosthesis for aortic arch diseases and found that the 30-day mortality after performing fenestrated TEVAR was 20%. Notably, aneurysms involving supra-aortic branches are not optimal candidates for the fenestration of the endoprosthesis due to the risk of endoleaks [3]. However, a previous dual-center study demonstrated acceptable outcomes of physician-modified thoracic stent-grafts with a single fenestration or a proximal scallop, with a 30-day mortality of 6% and aorta-related mortality of 6% [14]. Furthermore, Huang et al. [4] reported that type Ia endoleaks occurred in 17.8% of patients after zone 2 TEVAR using the chimney technique. In our study of zone 2 PMiBEVAR, no cases of in-hospital death, aneurysm diameter enlargement, or endoleak were observed. In addition, the hospital length of stay was shorter than other traditional techniques [15].

Custom-made endografts require 6–8 weeks for construction and delivery. Therefore, this technique cannot be used for emergency or urgent surgeries. Off-the-shelf grafts are immediately available and the procedure can be standardized. However, the grafts are not tailored to each patient, and additional procedures such as a bypass or a vascular plug are therefore required [16]. Previously, Bosse et al. [17] reported that five standardized off-the-shelf endografts could cover a majority of aortic arch anatomies. Rizza et al. [18] demonstrated that zone 2 TEVAR using a Castor single-branched stent graft manufactured by MicroPort Endovascular in Shanghai, China was safe because there were no perioperative mortalities, and no serious complications such as stroke, acute myocardial infarction, renal failure, or left arm ischemia occurred. In contrast, physician-modified endografts have definite anatomical suitability because they are created specifically for each individual patient. Therefore, the treatment of these patients can be completed using PMiBEVAR only. However, few previous studies have reported on PMiBEVAR for aortic arch aneurysm [19]. Notably, Zhang et al. [6] reported the successful application of a physician-modified endograft with triple inner branches for an zextensive aortic arch aneurysm. Postoperatively, they observed no endoleaks, and an aneurysm diameter shrinkage was shown during the 6-month follow-up [6]. Moreover, an increasing number of studies are reporting on PMiBEVAR for thoracoabdominal aortic aneurysms [20,21,22], showing that PMiBEVAR is a feasible approach for treating complex aneurysms [7].

Physician-modified stent-grafts can be attached at either the inner or outer branch; however, the inner branch is easier to cannulate from the LSCA than the outer branch. In addition, we believe that the landing length between the inner branch and the bridging stent-graft more effectively prevents type III endoleaks when compared with fenestration alone. Type III endoleaks from directional branches are a non-negligible complication after branched endovascular repair [23]. Another advantage of using the inner branch technique is that the graft is easier to resheath, which is an important factor for physician-modified stent-grafts. However, if type Ia endoleaks occur, reintervention is more difficult due to the presence of the inner branch. Thus, although PMiBEVAR is an effective treatment for aortic arch diseases, the complex structure makes this approach more difficult when complications occur. Therefore, the optimal reintervention methods for PMiBEVAR need to be evaluated [24,25].

As we accumulate more cases of PMiBEVAR, including that at zone 0 and zone 1, there is a possibility of improving techniques by introducing a stabilizing spine wire. Adding this wire is a crucial step in providing support and the secure alignment of the device and branch openings along the outer curve of the aortic arch [26]. In fact, modification techniques using a self-orienting spine trigger wire and anatomically specific fenestrations or inner branches have improved outcomes with regard to mortality, procedure time, and postoperative complications [27].

This study had some limitations. First, this retrospective analysis was conducted in a small population at a single center. Second, zone 0 and 1 PMiBEVAR outcomes were not evaluated. Third, long-term follow-up was not performed. Those limitations could affect and influence this study’s findings regarding the potential population bias and limited generalizability. Therefore, further large-scale and longer-term studies are required and it is necessary to compare the clinical outcomes of PMiBEVAR with established methods such as open surgery, hybrid procedures, and other techniques.

In conclusion, our study’s findings have demonstrated that PMiBEVAR for distal aortic arch aneurysm has acceptable short-term outcomes and may be effective in improving initial postoperative outcomes. However, further large-scale and longer-term studies are required to confirm the safety and efficacy of proximal landing zone 2 PMiBEVAR.

## Figures and Tables

**Figure 1 jcm-14-00039-f001:**
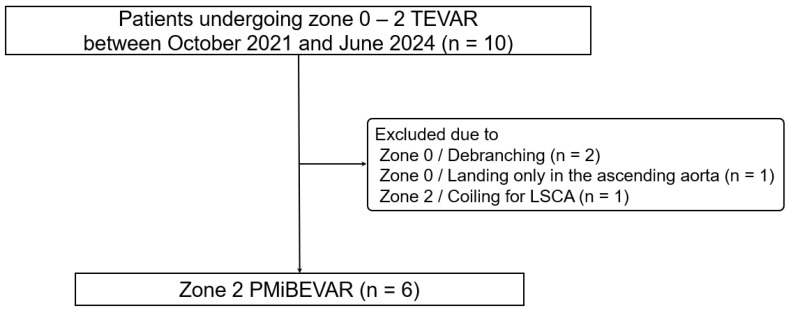
Study flow chart. TEVAR, thoracic endovascular aortic repair; LSCA, left subclavian artery; PMiBEVAR, physician-modified inner-branched endovascular repair.

**Figure 2 jcm-14-00039-f002:**
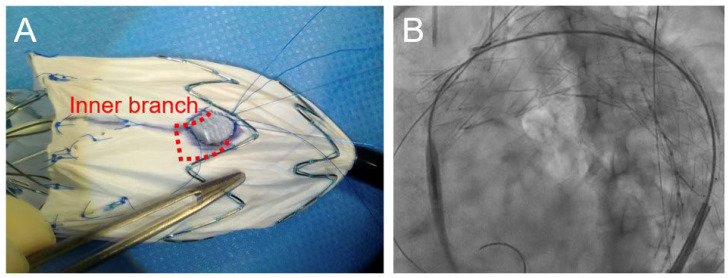
(**A**) Modifications of the main endograft device; and (**B**) intraoperative angiography showing the successful insertion of the bridging stent-graft.

**Figure 3 jcm-14-00039-f003:**
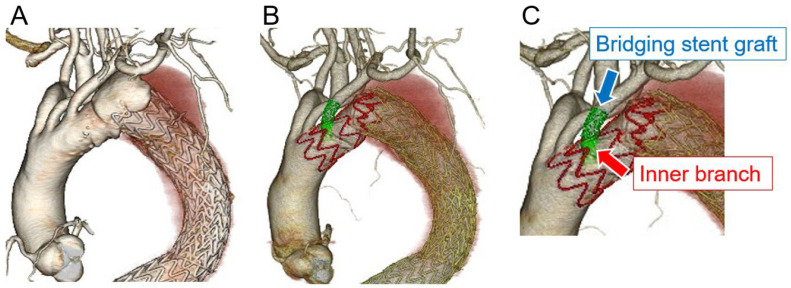
Contrast-enhanced computed tomography findings: (**A**) preoperative imaging; and (**B,C**) postoperative imaging.

**Table 1 jcm-14-00039-t001:** Preoperative characteristics.

Characteristics	Patients (*n* = 6)
Age, years	78.5 (76.5–79.0)
Sex, male	4 (66.7)
Cerebrovascular disease	3 (50.0)
CKD	3 (50.0)
COPD	3 (50.0)
Coronary artery disease	0 (0)
Diabetes mellitus	3 (50.0)
Dialysis-dependent CKD	1 (16.7)
Dyslipidemia	3 (50.0)
Hypertension	6 (100.0)
Aneurysm	
Aneurysm size, mm	56 (50–61)
True aneurysm	5 (83.3)
Pseudoaneurysm	0 (0)
Dissecting aneurysm	1 (16.7)
ASA class	
II	1 (16.7)
III	4 (66.7)
IV	1 (16.7)
Previous aortic surgery	3 (50.0)

Data are presented as *n* (%) or medians (interquartile ranges). Abbreviations: ASA, American Society of Anesthesiologists; CKD, chronic kidney disease; COPD, chronic obstructive pulmonary disease.

**Table 2 jcm-14-00039-t002:** Procedural details.

Procedure	Patients (*n* = 6)
Blood transfusions	4 (66.7)
Contrast volume, mL	103 (80–144)
Modification time, min	56 (45–60)
Devices	
Modified device	6 (100.0)
Relay^®^Plus ^1^	1 (16.7)
Zenith Alpha™ ^2^	5 (83.3)
Inner branch device	6 (100.0)
Viabahn^® 3^	6 (100.0)
Bridge stent device	5 (83.3)
LifeStream™ ^4^	2 (33.3)
Viabahn^®^ VBX ^5^	1 (16.7)
Viabahn^® 3^	2 (33.3)
Estimated blood loss, mL	50 (49–840)
Fluoroscopy time, min	33 (24–138)
Operative time, min	92 (79–308)
Percutaneous approach	6 (100.0)
Technical success	5 (83.3)
Total radiation dose, mGy	367 (341–1845)

Data are presented as *n* (%) or medians (interquartile ranges). ^1^ Relay^®^Plus stent-graft (Terumo Aortic, Sunrise, FL, USA). ^2^ Zenith Alpha™ thoracic stent-graft (Cook Medical, Bloomington, IN, USA). ^3^ Viabahn^®^ stent-graft (WL Gore & Associates, Flagstaff, AZ, USA). ^4^ LifeStream™ (Becton, Dickinson and Bard Company, Tempe, AZ, USA). ^5^ Viabahn^®^ VBX (WL Gore & Associates, Flagstaff, AZ, USA).

**Table 3 jcm-14-00039-t003:** Devices of all patients.

Patient	Modified Device	Modified Device Size, mm	Delivery System	Inner Branch Device	Bridge Stent Device	Bridge Stent Device Size, mm
1	Relay^®^Plus ^1^	34	24 Fr	Viabahn^® 3^		
2	Zenith Alpha™ ^2^	42	21 Fr	Viabahn^® 3^	Viabahn^® 3^	13
3	Zenith Alpha™ ^2^	34	21 Fr	Viabahn^® 3^	LifeStream™ ^4^	9
4	Zenith Alpha™ ^2^	32	20 Fr	Viabahn^® 3^	Viabahn^®^ VBX ^5^	8
5	Zenith Alpha™ ^2^	34	20 Fr	Viabahn^® 3^	LifeStream™ ^4^	9
6	Zenith Alpha™ ^2^	44	21 Fr	Viabahn^® 3^	Viabahn^® 3^	13

^1^ Relay^®^Plus stent-graft (Terumo Aortic, Sunrise, FL, USA). ^2^ Zenith Alpha™ thoracic stent-graft (Cook Medical, Bloomington, IN, USA). ^3^ Viabahn^®^ stent-graft (WL Gore & Associates, Flagstaff, AZ, USA). ^4^ LifeStream™ (Becton, Dickinson and Bard Company, Tempe, AZ, USA). ^5^ Viabahn^®^ VBX (WL Gore & Associates, Flagstaff, AZ, USA).

**Table 4 jcm-14-00039-t004:** Postoperative outcomes.

Outcomes	Patients (*n* = 6)
Acute renal failure	0 (0)
Brain infarction	0 (0)
Endoleak (delayed type Ia)	1 (16.7)
Postoperative length of stay, days	10 (7–41)
Reintervention	0 (0)
Spinal cord ischemia	0 (0)
Vascular access-site complications ^1^	1 (16.7)
In-hospital mortality	0 (0)

Data are presented as *n* (%) or medians (interquartile ranges). ^1^ The one case of vascular access-site complications was a pseudoaneurysm.

**Table 5 jcm-14-00039-t005:** Outcomes during post-discharge follow-up.

Outcomes	Patients (*n* = 6)
Aortic remodeling	
Enlargement	0 (0)
No change	4 (66.7)
Shrinkage	2 (33.3)
Brain infarction	0 (0)
Endoleak	0 (0)
Reintervention	0 (0)
Renal failure	0 (0)
Spinal cord ischemia	0 (0)
Follow-up period, months	12.5 (7.3–25)

Data are presented as *n* (%) or medians (interquartile ranges).

**Table 6 jcm-14-00039-t006:** Details of all patients.

Patient	Modification Time, min	Operative Time, min	Technical Success	Re- Intervention	In-Hospital Death	Postoperative Length of Stay, Days	Aortic Remodeling
1	60	807	No	No	No	99	No change
2	60	92	Yes	No	No	8	No change
3	60	92	Yes	No	No	12	Shrinkage
4	40	59	Yes	No	No	7	Shrinkage
5	46	85	Yes	No	No	21	No change
6	52	142	Yes	No	No	7	No change

## Data Availability

The original data presented in the study are openly available from corresponding author.

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
