# Peer review of "Initial Outcomes of Physician-Modified Inner-Branched Endovascular Repair for Distal Aortic Arch Aneurysm"

_jcm, 2024, doi:10.3390/jcm14010039_

Round 1
Reviewer 1 Report
Comments and Suggestions for Authors
We express our gratitude to the authors for submitting their manuscript to our journal. The article focuses on the "Initial Outcomes of Physician-Modified Inner-Branched Endovascular Repair for Distal Aortic Arch Aneurysm." The results are presented optimally and are indeed intriguing. However, we request minor revisions to enhance the clarity and depth of the study.
Firstly, we recommend the creation of a graphical abstract that succinctly summarizes the study's main results and key messages. This visual representation would significantly aid in the comprehension of the findings.
Secondly, it is suggested that the authors discuss their results in comparison with those presented by Rizza et al. in their study on "Preliminary Outcomes of Zone 2 Thoracic Endovascular Aortic Repair Using Castor Single-Branched Stent Grafts," (PMID: 38137662, PMCID: PMC10743804, DOI: 10.3390/jcm12247593)
Additionally, we advise the inclusion of a flow chart outlining the patient selection process, which would enhance the methodology section by clearly illustrating the study design.
Lastly, we encourage the authors to expand the introduction and discussion to incorporate references to other endovascular techniques utilized for Left Subclavian Artery (LSCA) revascularization within the framework of Thoracic Endovascular Aortic Repair.
Author Response
We express our gratitude to the authors for submitting their manuscript to our journal. The article focuses on the "Initial Outcomes of Physician-Modified Inner-Branched Endovascular Repair for Distal Aortic Arch Aneurysm." The results are presented optimally and are indeed intriguing. However, we request minor revisions to enhance the clarity and depth of the study.
Firstly, we recommend the creation of a graphical abstract that succinctly summarizes the study's main results and key messages. This visual representation would significantly aid in the comprehension of the findings.
Response: Thank you for this important comment. Per your comment, we have added the graphical abstract.
Secondly, it is suggested that the authors discuss their results in comparison with those presented by Rizza et al. in their study on "Preliminary Outcomes of Zone 2 Thoracic Endovascular Aortic Repair Using Castor Single-Branched Stent Grafts," (PMID: 38137662, PMCID: PMC10743804, DOI: 10.3390/jcm12247593)
Response: Thank you for your suggestion. Per your comment, we have added this information to the revised manuscript as follows:
“Rizza et al. [18] demonstrated that zone 2 TEVAR using a Castor single-branched stent graft manufactured by MicroPort Endovascular in Shanghai, China was safe because there were no perioperative mortalities and no serious complications such as stroke, acute myocardial infarction, renal failure, or left arm ischemia occurred.” (P7–8 L218–222)
Additionally, we advise the inclusion of a flow chart outlining the patient selection process, which would enhance the methodology section by clearly illustrating the study design.
Response: Thank you for your suggestion. Per your comment, we have added this information to Figure 1.
Lastly, we encourage the authors to expand the introduction and discussion to incorporate references to other endovascular techniques utilized for Left Subclavian Artery (LSCA) revascularization within the framework of Thoracic Endovascular Aortic Repair.
Response: Thank you for your suggestion. Per your comment, we have added this information to the revised manuscript as follows:
“Although frequently requiring debranching to maintain patency of LSCA, endovascular arch repair is an attractive alternative procedure for high surgical risk patients. Fenestrated endograft repair of aortic arch disease has a high technical success rate (95%), however, a high number of secondary interventions were needed [5]. On the other hand, zone 2 TEVAR using the chimney technique has a high risk of type Ia endoleaks [4].” (P1 L39–44)
“In our study of zone 2 PMiBEVAR, no cases of in-hospital death, aneurysm diameter enlargement or endoleak were observed. In addition, hospital length of stay was shorter than other traditional techniques [15].” (P7 L210–212)
Reviewer 2 Report
Comments and Suggestions for Authors
The authors represent a retrospective analysis of six patients who underwent zone 2 PMiBEVAR for aortic arch aneurysms at a single center between October 2021 and June 2024. With a manuscript title "Initial outcomes of physician-modified inner-branched endovascular repair for distal aortic arch aneurysm".
ABSTRACT
Results
Please mention the gender of the studied patients, i.e. how many males and how many females.
The authors mentioned that: the median aneurysm diameter was 56 (50–61) mm. As mentioned, the lower aneurysm diameter was 50 mm; did you operate on this diameter? We operate if the aneurysm diameter is 55 mm or more, (please clarify).
The authors mentioned that: vascular access-site complication occurred in one patient. Please mention the type of reported complication. Is it access site hematoma, pseudoaneurysm, AV fistula etc... as it affects only one patient?
MAIN MANUSCRIPT
Introduction
Lines 38-40: At our institution, we perform physician-modified inner-branched endovascular repair (PMiBEVAR) in these cases, using an inner-branch and bridging stent-graft to ensure perfusion of the LSCA. Please remove this sentence from the introduction to be preferably placed in the methodology section of the main manuscript.
Materials and Methods
Lines 51-52: This retrospective study analyzed data from six patients who underwent zone 2 51PMiBEVAR for distal aortic arch aneurysms. Please specify the distally located aneurysm as regards its relation to the left subclavian artery.
Line 65: Please mention the type of imaging modalities performed for the study.
Line 77: Please mention the mode of sterilization after modification of the stent graft and manual re-sheathing.
Lines 88-89: We routinely performed postoperative CT. Is it right to mention only CT or you should mention postoperative CTA. Please correct.
Author Response
The authors represent a retrospective analysis of six patients who underwent zone 2 PMiBEVAR for aortic arch aneurysms at a single center between October 2021 and June 2024. With a manuscript title "Initial outcomes of physician-modified inner-branched endovascular repair for distal aortic arch aneurysm".
ABSTRACT
Results
Please mention the gender of the studied patients, i.e. how many males and how many females.
Response: Thank you for your suggestion. Per your comment, we have added this information to the revised manuscript as follows:
“Males were 4/6 and females were 2/6 cases.” (P1 L16–17)
The authors mentioned that: the median aneurysm diameter was 56 (50–61) mm. As mentioned, the lower aneurysm diameter was 50 mm; did you operate on this diameter? We operate if the aneurysm diameter is 55 mm or more, (please clarify).
Response: Per your comment, we usually operated on 55 mm or greater, but those cases with the aneurysm diameter less than 55 mm were saccular aneurysms with a high risk of rupture.
The authors mentioned that: vascular access-site complication occurred in one patient. Please mention the type of reported complication. Is it access site hematoma, pseudoaneurysm, AV fistula etc... as it affects only one patient?
Response: Per your comment, we have added this information to the revised manuscript as follows:
“vascular access-site pseudoaneurysm occurred in one patient each.” (P1 L20)
MAIN MANUSCRIPT
Introduction
Lines 38-40: At our institution, we perform physician-modified inner-branched endovascular repair (PMiBEVAR) in these cases, using an inner-branch and bridging stent-graft to ensure perfusion of the LSCA. Please remove this sentence from the introduction to be preferably placed in the methodology section of the main manuscript.
Response: Per your comment, we have revised the manuscript as follows:
“At our institution, we perform PMiBEVAR, using an inner branch and bridging stent-graft to ensure perfusion of the LSCA.” (P2 L56–57)
Materials and Methods
Lines 51-52: This retrospective study analyzed data from six patients who underwent zone 2 51PMiBEVAR for distal aortic arch aneurysms. Please specify the distally located aneurysm as regards its relation to the left subclavian artery.
Response: Thank you for your suggestion. Per your comment, we have added this information to the revised manuscript as follows:
“a proximal landing zone (the distance between the proximal end and the ostium of the LSCA) of at least 15 mm for distal aortic arch aneurysm.” (P2 L69–71)
Line 65: Please mention the type of imaging modalities performed for the study.
Response: Per your comment, we have added this information to the revised manuscript as follows:
“(computed tomography; CT)” (P2 L72)
Line 77: Please mention the mode of sterilization after modification of the stent graft and manual re-sheathing.
Response: Thank you for your comment. Since the stent graft was created sterile procedures, no sterilization was performed after re-sheathing.
Lines 88-89: We routinely performed postoperative CT. Is it right to mention only CT or you should mention postoperative CTA. Please correct.
Response: Thank you for your comment. “We routinely performed postoperative CTA.” (P3 L99)
Reviewer 3 Report
Comments and Suggestions for Authors
I had the opportunity of reviewing the manuscript titled "Initial Outcomes of Physician-Modified Inner-Branched Endovascular Repair for Distal Aortic Arch Aneurysm”.
The main aim of this retrospective study was to investigate the initial outcome of physician-modified inner branched endovascular repair (PMiBEVAR) for zone 2 aortic arch aneurysm. The outcome parameters analyzed in this study were in-hospital mortality and postoperative 15 complications.
Overall, I found the article well written, and the topic is relevant to the medical community, especially considering the growing interest in physician-modified endografts.
Analyzing the specific section of the manuscript, I’d like to offer the following feedback to enhance the overall quality further.
Introduction section:
- I think that the authors could give a better comprehensive presentation of the PMiBEVAR and how and why this technique have gained a role in zone 2 aortic arch aneurysm repair by citing more relevant literature
Materials and Methods section:
- The authors should give more detailed explanation, by citing relevant literature or reorganizing the paragraph on how they decided to determine the parameters used to define “technical success” and “postoperative complications” (with a thorough explanation of the origin of the composite endpoint), and clarify the scientific basis (citing relevant literature) upon which the reintervention criteria were established.
Narrative review section:
- I would move paragraphs 171 to 176 to the beginning of section 3.2, as it would be clearer for the readers and make the discussion more cohesive
- Line 259-260, please insert significant citations from the literature
Discussion section:
- The authors have correctly mentioned the advantages of PMiBEVAR over other more traditional tecniques like debranching or chimney techinques, but I would add a more in-depth analysis of the cited literature in terms of other outcomes such as morbidity or complications, length of stay. This argumentation would increase the overall argument for PMiBEVAR's feasibility.
- In the limitations section, the authors correctly acknowledged the limited study population and the retrospective nature of the study. However I think that the manuscript could benefit from a better discussion about how this limitation could affect and influence the study’s findings regarding potential population bias and limited generalizability.
Author Response
I had the opportunity of reviewing the manuscript titled "Initial Outcomes of Physician-Modified Inner-Branched Endovascular Repair for Distal Aortic Arch Aneurysm”.
The main aim of this retrospective study was to investigate the initial outcome of physician-modified inner branched endovascular repair (PMiBEVAR) for zone 2 aortic arch aneurysm. The outcome parameters analyzed in this study were in-hospital mortality and postoperative 15 complications.
Overall, I found the article well written, and the topic is relevant to the medical community, especially considering the growing interest in physician-modified endografts.
Analyzing the specific section of the manuscript, I’d like to offer the following feedback to enhance the overall quality further.
Introduction section:
- I think that the authors could give a better comprehensive presentation of the PMiBEVAR and how and why this technique have gained a role in zone 2 aortic arch aneurysm repair by citing more relevant literature
Response: Thank you for your suggestion. Per your comment, we have added this information to the revised manuscript as follows:
“Although frequently requiring debranching to maintain patency of LSCA, endovascular arch repair is an attractive alternative procedure for high surgical risk patients. Fenestrated endograft repair of aortic arch disease has a high technical success rate (95%), however, a high number of secondary interventions were needed [5]. On the other hand, zone 2 TEVAR using the chimney technique has a high risk of type Ia endoleaks [4].” (P1 L39–44)
Materials and Methods section:
- The authors should give more detailed explanation, by citing relevant literature or reorganizing the paragraph on how they decided to determine the parameters used to define “technical success” and “postoperative complications” (with a thorough explanation of the origin of the composite endpoint), and clarify the scientific basis (citing relevant literature) upon which the reintervention criteria were established.
Response: Thank you for your suggestion. Per your comment, we have added this information to the revised manuscript as follows:
“Decisions regarding reintervention were made based on the following factors: aneurysm growth of >5 mm when compared with the preoperative measurement, the presence or absence of type I or type III endoleaks, graft migration, graft occlusion, rupture, and infection following the guideline [9].” (P3 L101–104)
“Imitating other PMiBEVAR study [7], technical success was defined on an intention-to-treat basis as successful endovascular access and deployment of all devices, successful catheterization and stenting of the LSCA, no target vessel occlusion on control angiogram. Postoperative complications were defined using a composite endpoint following previous reports [2, 4], including all-cause mortality, endoleak, acute renal failure (>50% decrease in estimated glomerular filtration rate), spinal cord ischemia (non-ambulatory, Spinal Cord Injury scale grade 3a–3c) [10], and stroke.” (P2–3 L92–98)
Narrative review section:
- I would move paragraphs 171 to 176 to the beginning of section 3.2, as it would be clearer for the readers and make the discussion more cohesive
- Line 259-260, please insert significant citations from the literature
Response: Thank you for your suggestion. Per your comment, we have moved paragraphs to the beginning of section 3.2. Moreover, we have inserted significant citations to the revised manuscript as follows:
“The endovascular and hybrid repair of the aortic arch has seen a rise in novel technologies and the literature over the last decade [1].” (P2 L32–34)
Discussion section:
- The authors have correctly mentioned the advantages of PMiBEVAR over other more traditional tecniques like debranching or chimney techinques, but I would add a more in-depth analysis of the cited literature in terms of other outcomes such as morbidity or complications, length of stay. This argumentation would increase the overall argument for PMiBEVAR's feasibility.
- In the limitations section, the authors correctly acknowledged the limited study population and the retrospective nature of the study. However I think that the manuscript could benefit from a better discussion about how this limitation could affect and influence the study’s findings regarding potential population bias and limited generalizability.
Response: Thank you for your suggestion. Per your comment, we have added the advantages of PMiBEVAR as follows:
“In our study of zone 2 PMiBEVAR, no cases of in-hospital death, aneurysm diameter enlargement or endoleak were observed. In addition, hospital length of stay was shorter than other traditional techniques [15].” (P7 L210–212)
Moreover, we have revised the limitation section as follows:
“Those limitations could affect and influence this study findings regarding potential population bias and limited generalizability. Therefore, further large-scale and longer-term studies are required and it is necessary to compare the clinical outcomes of PMiBEVAR with established methods such as open surgery, hybrid procedures, and other techniques.” (P8 L253–257)
Round 2
Reviewer 3 Report
Comments and Suggestions for Authors
In the revised version of the manuscript, the authors successfully addressed all the concerns raised in the previous review. The introduction is now clearer, with relevant literature and the advantages of PMiBEVAR are clearly stated, but more importantly the definition of technical success is now clearer for the reader. Also the concerns in the limitations and discussion sections were amended correctly, increasing the overall value of the article despite making clear the retrospective nature and limited sample size of the findings.